# Combinatory Treatment of Canavanine and Arginine Deprivation Efficiently Targets Human Glioblastoma Cells via Pleiotropic Mechanisms

**DOI:** 10.3390/cells9102217

**Published:** 2020-09-30

**Authors:** Olena Karatsai, Pavel Shliaha, Ole N. Jensen, Oleh Stasyk, Maria Jolanta Rędowicz

**Affiliations:** 1Nencki Institute of Experimental Biology, Polish Academy of Sciences, 3 Pasteur St., 02-093 Warsaw, Poland; o.karatsai@nencki.edu.pl; 2Department of Biochemistry and Molecular Biology and VILLUM Center for Bioanalytical Sciences, University of Southern Denmark, DK-5230 Odense M, Denmark; shliahap@mskcc.org (P.S.); jenseno@bmb.sdu.dk (O.N.J.); 3Department of Cell Signaling, Institute of Cell Biology, National Academy of Sciences of Ukraine, 14/16 Drahomanov St., 79005 Lviv, Ukraine; oleh11@gmail.com

**Keywords:** adhesion, apoptosis, arginine deprivation, canavanine, cancer, cytoskeleton, ER stress, glioblastoma, migration, unfolded protein response

## Abstract

Glioblastomas are the most frequent and aggressive form of primary brain tumors with no efficient cure. However, they often exhibit specific metabolic shifts that include deficiency in the biosynthesis of and dependence on certain exogenous amino acids. Here, we evaluated, in vitro, a novel combinatory antiglioblastoma approach based on arginine deprivation and canavanine, an arginine analogue of plant origin, using two human glioblastoma cell models, U251MG and U87MG. The combinatory treatment profoundly affected cell viability, morphology, motility and adhesion, destabilizing the cytoskeleton and mitochondrial network, and induced apoptotic cell death. Importantly, the effects were selective toward glioblastoma cells, as they were not pronounced for primary rat glial cells. At the molecular level, canavanine inhibited prosurvival kinases such as FAK, Akt and AMPK. Its effects on protein synthesis and stress response pathways were more complex and dependent on exposure time. We directly observed canavanine incorporation into nascent proteins by using quantitative proteomics. Although canavanine in the absence of arginine readily incorporated into polypeptides, no motif preference for such incorporation was observed. Our findings provide a strong rationale for further developing the proposed modality based on canavanine and arginine deprivation as a potential antiglioblastoma metabolic therapy independent of the blood–brain barrier.

## 1. Introduction

It has been described in a number reports that numerous cancers are defective in arginine biosynthesis, and some of them become hypersensitive to the deprivation of this amino acid [1]. Arginine is a semi-essential amino acid in humans, and its exogenous requirement is dependent on the stage of the organism’s development and its health status [2,3]. Arginine plays an important role in various molecular pathways regulating cell division, wound healing, neurological and immune functions, and hormone synthesis [1,4,5]. It is also a key precursor in the synthesis of cancer-associated compounds such as nitric oxide (NO) and polyamines [3,6,7].

In the normal condition, intracellular arginine homeostasis depends on dietary uptake, the degradation of intracellular proteins, efficiency of transport through cell membranes by cationic amino acid transporters (CATs), and de novo synthesis from aspartate and citrulline in the urea cycle [7,8,9]. There are two closely coupled enzymes, which are crucial in arginine biosynthesis: argininosuccinate synthetase 1 (ASS1) and argininosuccinate lyase (ASL) [4]. The auxotrophy for arginine, in vivo, of several tumors, such as melanoma, hepatocellular carcinoma, some mesotheliomas, some renal cell cancers, and glioblastomas, is mainly related to the epigenetic silencing of ASS1 [9,10,11]. In contrast to cancer cells with elevated nutrient requirements, the majority of normal non-transformed cells are able to synthesize arginine by conversion from circulating exogenous citrulline mediated by ASS and ASL under conditions of nutrient stress [12].

Arginine deprivation with recombinant arginine-degrading enzymes is considered as nontoxic and selective since it decreases viability and allows the control of the growth of malignant tumors that lack ASS1 expression [4,5,13,14,15]. Such a strategy is now under Phase I/II clinical trials for combating malignant melanomas, hepatocellular and pancreatic carcinomas, pleural mesotheliomas, thoracic cancers and other malignancies, and is well tolerated by the organism [16,17,18,19,20,21,22].

Due to specific metabolic shifts in subsets of glioblastomas, the strategy of arginine deprivation could also be a promising approach for the treatment of these devastating cancers [7,9,23,24]. It was observed by several groups, including ours, that arginine deprivation affects glioblastoma cell viability, morphology and invasiveness [1,25]. However, the effects of arginine deprivation on glioblastoma cells were reversible after arginine resupplementation. Three-hour culture in arginine-enriched conditions leads to the restoration of cancer cell morphology and growth [25]. In the majority of tumors, arginine deprivation affects only cell growth and does not promote cell death, therefore bringing the danger of cancer recurrence after stopping the treatment. To overcome this problem, we propose combining arginine deprivation-based treatment with an arginine antimetabolite, l-canavanine.

l-canavanine, a non-proteinogenic amino acid found in certain leguminous plants, is a naturally occurring structural analog of l-arginine. The sole structural difference between arginine and canavanine is the replacement of carbon with oxygen in the terminal methylene group (Figure 1) [26]. There are several reports that consider canavanine as a possible proteomimetic amino acid that could be incorporated instead of arginine into newly synthesized proteins [13,27]. Importantly, in the case of gliomas as targets, the effect of the combination of enzymatic arginine deprivation with canavanine does not depend on the permeability of the brain–blood barrier, as canavanine is transported by the same carriers as arginine, thus increasing the feasibility of a potential prospective therapy [27].

In the current study, we examined, in detail, the combined effects of canavanine in arginine-free conditions on the proliferation, morphology, motility and adhesion of two human glioblastoma cell lines. Primary rat glial cells served as control non-transformed cells. We observed that under arginine deprivation, canavanine stimulated pleiotropic mechanisms leading to apoptotic cell death only in cancer but not in normal cells. These data support the notion that canavanine is a good potential candidate in the development of combination metabolic antiglioblastoma therapy.

## 2. Materials and Methods

### 2.1. Materials

l-canavanine (Sigma-Aldrich C9758, St. Louis, MO, USA) was dissolved in water to a 50 mM concentration and stored at 4 °C. The MTS reagent (CellTiter 96^®^ AQueous Non-Radioactive Cell Proliferation Assay, G5421) was from Promega Corporation (Madison, WI, USA). ^13^C_6_,^15^N_4_-arginine was obtained from ThermoFisher Scientific (89990, Waltham, MA, USA). ER-Tracker™ Blue-White DPX was obtained from ThermoFisher Scientific (E12353). Alexa Fluor 488- and 546-conjugated phalloidin were obtained from Invitrogen (A12379 and A22283, respectively; Waltham, MA, USA). Vectashield anti-fade reagent with or without DAPI was obtained from Vector Laboratories (H-1200 and H-1000, respectively; Burlingame, CA, USA). The following antibodies were used: those against c-PARP (Cell Signaling Technologies #9546, Danvers, MA, USA, and Enzo Life Sciences BML-SA249-0100, Farmingdale, NY, USA), caspase 3 (#9662) and caspase 9 (#9502), FAK (#3285) and p-FAK (#8556), eIF2α (#2103) and p-eIF2α (#9721), SAPK/JNK (#9252) and p-SAPK/JNK (#4668), AMPKα (#2532) and p-AMPKα (#2535), p38 (#9212) and p-p38 (#9211), S6 (#2217) and p-S6 (#4858), 4EBP1 (#9452), ATF-4 (#11815), CHOP (#2895), lamin A/C (#4777), tensin 2 (#11990), LC3A/B (#4108) (Cell Signaling Technologies), talin (sc-7534), HSP60 (sc-59567), HSP70 (sc-32239) (Santa Cruz Biotechnology, Heidelberg, Germany), vinculin and β-tubulin (V4505 and T0198, respectively; Sigma-Aldrich), GRP78 BiP (ab21685), GFAP (ab7260), GRP75/MOT (ab2799), lamin B1 (ab16048) and lamin B2 (ab151735) (Abcam, Cambridge, UK), and GAPDH (Millipore MAB374, St. Louis, MO, USA). The following secondary antibodies were used: HRP-conjugated anti-mouse and anti-rabbit IgG (AP308P and AP307P, respectively; Millipore), HRP-conjugated anti-goat IgG (Santa Cruz Biotechnology sc-2020), Alexa Fluor 546-conjugated anti-mouse, Alexa Fluor 488-conjugated anti-rabbit (A11003 and A11008, respectively; Millipore) and Alexa Fluor 488-conjugated anti-mouse (Abcam ab150113). The protein bands were visualized using ECL reagent (Millipore P90720).

### 2.2. Cell Lines and Culture Conditions

Human glioblastoma U251MG and U87MG cell lines were obtained from the Cell Lines Service (Eppelheim, Germany). Cells were cultivated at 37 °C with 5% CO_2_ in Dulbecco’s Modified Eagle Medium (DMEM; Gibco 31966021, Waltham, MA, USA) supplemented with GlutaMAX-1, 10% heat inactivated fetal bovine serum (FBS; Gibco 10500064) and antibiotics, 1% penicillin/streptomycin (Gibco 15140122). For all experiments, we formulated DMEM (Sigma-Aldrich D9443) with or without 0.4 mmol/L arginine (complete medium (CM) and arginine-free medium (AFM), respectively), supplemented with 5% dialyzed serum (Sigma-Aldrich F0392).

### 2.3. Primary Rat Glial Cell Culture

The primary culture of rat glial cells, used here as the control non-modified normal cells, was isolated from one-day-old Wistar rat pups as described by [28] with the following modifications. The forebrains were isolated, washed with ice-cold DMEM GlutaMAX-1 supplemented with 1% penicillin/streptomycin, and transferred to 0.25% trypsin-EDTA (Gibco 25200056) for 25 min at 37 °C with 5% CO_2_. Next, DMEM, supplemented with 10% FBS and antibiotics, was added to the tissue suspension, followed by mechanical dissociation by pipetting, and the suspension was passed through a 70 µm cell strainer (Falcon^®^ 352350, Glendale, AZ, USA). The single cells were plated into poly-L-lysine-coated (100 µg/mL; Sigma-Aldrich P4707) flasks and maintained in complete DMEM. After 3 days, 2/3 of the medium was refreshed. On Day 7, the flasks with primary glial cultures were transferred to an orbital shaker at 100 rpm for 30 min to remove microglia. After that, the entire medium with detached cells was removed. The homogeneity of the cell population was maintained for at least four passages. More than 80% of the cells were astrocytes as confirmed by immunofluorescence staining for glial fibrillary acidic protein (GFAP). Afterwards, cells were seeded and cultured under experimental conditions.

### 2.4. MTS Cell Viability Assay

The cell viability MTS assay was performed according to the manufacturer’s instructions. Briefly, cells were seeded in a regular DMEM medium in 96-well plates (8 × 10^3^ cells/well) and allowed to adhere for 18 h, and then, the medium was removed. Next, the cells were washed twice with phosphate buffered saline (PBS) and finally treated with different canavanine concentrations (10–250 μM) in complete medium or arginine-free medium for 24, 48 and 72 h. MTS solution was added to each well, and the cells were further incubated for 1 h. The absorbance was measured using a Tecan SUNRISE XFluor4 plate reader at a wavelength of 490 nm. The percentage of viable cells after treatment was calculated by assuming 100% viability for the absorbance recorded for the control conditions (i.e., in the absence of canavanine).

### 2.5. Western Blot Analysis

The cells were washed three times in ice-cold PBS and lysed in RIPA Buffer (50 mM Tris-HCl pH 7.5, 150 mM NaCl, 0.5% sodium deoxycholate, 0.1% SDS, 1% IGEPAL (nonionic, non-denaturing detergent), 50 mM NaF, 2 mM Na_3_VO_4_, 1 mM PMSF, and protease and phosphatase inhibitor cocktails (04693116001 and 04906837001, respectively; Roche, Mannheim, Germany) at 4 °C for 20 min. Cell extracts were obtained after centrifugation at 13,500× *g* at 4 °C for 20 min, and the supernatants were used for the study. Protein concentrations were quantified according to the Bradford method. Then, the supernatants were incubated with the Laemmli buffer for 5 min at 98 °C. Equal amounts of protein were separated in 10, 12 or 15% SDS–polyacrylamide gels (SDS-PAGE) and transferred onto nitrocellulose membranes (Amersham 10600002, Freiburg, Germany). The membranes were blocked with 3–5% fat-free milk or 5% BSA (Bioshop ALB001, Burlington, Canada) in TBS containing 0.2% Triton X-100 and then probed with the appropriate primary and secondary antibodies. β-Tubulin and GAPDH were used as protein loading controls. After 2 h, one-minute-UV-irradiated Jurkat cells were lysed and were subjected to analysis as an apoptotic cell death positive control. The protein bands were visualized using ECL reagent. Band densitometry quantification was performed using the Fiji distribution of the ImageJ 1.52a software (National Institutes of Health and the University of Wisconsin, Madison, WI, USA).

### 2.6. Immunocytochemical Staining and Microscopy Analysis

Cells were seeded on glass coverslips (VWR 631-0153, Gdańsk, Poland), cultured in respective conditions and then washed twice with PBS, fixed with 4% paraformaldehyde solution (PFA) for 20 min, washed twice with PBS, quenched for 30 min with 50 mM NH_4_Cl, permeabilized with 0.2% Triton X-100 in PBS for 10 min and finally incubated for 1.5 h in a blocking solution (2% horse serum in PBS/0.02% Triton X-100). To visualize actin filaments, cells were stained for 20 min with Alexa Fluor 488- or 546-conjugated phalloidin (diluted 1:40 in PBS) and then washed three times with PBS/0.02% Triton X-100. Next, cells were incubated overnight at 4 °C with primary antibodies and for 1.5 h at room temperature with Alexa Fluor 546- or 488-conjugated anti-mouse or Alexa Fluor 488-conjugated anti-rabbit antibodies diluted to 1:1000. Coverslips were extensively washed in PBS/0.02% Triton X-100 and mounted using Vectashield anti-fade reagent supplemented with DAPI to stain nuclei. Images were taken with the Zeiss LSM780, Inverted Axio Observer Z.1 with Plan Apochromat 40×/1.4 and 63×/1.4 Oil DIC objectives. The images were processed using the Zen Blue 2.1 software (Carl Zeiss Microscopy, Jena, Germany).

### 2.7. Confocal Endoplasmic Reticulum Localization

The endoplasmic reticulum was visualized by staining with the endoplasmic reticulum (ER)-specific dye, ER Tracker™ Blue/White DPX, which is retained within the ER lumen, thus labeling the ER tubular network. The assay was performed according to the manufacturer’s instructions. Briefly, cells were seeded on the glass coverslips and cultured under respective conditions. After that, the cells were incubated for 30 min at 37 °C and 5% CO_2_ with 1 µM ER Tracker diluted in experimental conditions. Then, the stained cells were fixed with 4% formaldehyde for 10 min, washed and mounted using the Vectashield anti-fade reagent. Images were taken with the Zeiss LSM780, Inverted Axio Observer Z.1 with Plan Apochromat 63×/1.4 Oil DIC objectives. The images were processed using the Zen Blue 2.1 software (Carl Zeiss Microscopy).

### 2.8. Transwell Migration Assay

Cell migration was assayed using 24-well Transwell^TM^ chambers with 6.5 mm-diameter polycarbonate filters with 8 μm-pore-size Transwell^TM^ migration inserts according to the manufacturer’s instructions with the following modifications (Corning CLS3422-48EA, Glendale, AZ, USA). The assay was based on chemotactic directional migration. Cells were treated under experimental conditions for 48 h, as were the control cells. Then, the cells were trypsinized, washed twice with serum-free medium and counted. The same numbers of living cells were seeded (2.5−3 × 10^4^ cells/well depending on the cell line) on the upper chamber and allowed to migrate through the filter to the lower chamber containing DMEM with 10% FBS. After 8–18 h (for U251MG and U87MG cells, respectively), the cells were fixed with 4% PFA for 15 min and stained with 0.5% crystal violet (Sigma-Aldrich) for 7–10 min. Cells that did not migrate through the filter were removed from the upper chamber by using a cotton swab. Migrated cells were photographed using a Nikon Eclipse Ti-U fluorescent microscope equipped with a 20× objective and DS-Qi2 digital camera and then counted with the Fiji distribution of the ImageJ 1.52a software (National Institutes of Health and the University of Wisconsin). The percentage of migrated cells was calculated by assuming 100% for the control conditions.

### 2.9. Cell Adhesion Assay

The cell adhesion assay was performed with the CytoSelect^TM^ 48-well cell adhesion assay according to the instructions provided by the manufacturer (Cell Biolabs, Inc. CBA-070, San Diego, CA, USA). Briefly, cells were cultured under the indicated experimental conditions for 48 h as were the control cells, trypsinized, washed once with medium with serum and twice with serum-free medium, and then seeded (10^5^ living cells/well) onto the collagen I-coated wells. After 2.5 h, unbound cells were washed away with PBS, and the adherent cells were stained. Finally, the stain was extracted and the OD560 measured with a Tecan SUNRISE XFluor4 plate reader.

### 2.10. Stable Isotope Labeling by Amino Acids in Cell Culture (SILAC)

Amino acid labeling was performed as described by Ong and Mann [29]. We performed the experiments on U251MG cells, for which the effects of both arginine deprivation and canavanine co-treatment were more visible. The cells were grown to 80–90% confluency in standard DMEM medium formulated with heavy ^13^C_6_,^15^N_4_-arginine (0.1 mM) in culture dishes. The medium was changed every 2–3 days in the cases when the cells were not ready for passaging.

### 2.11. Protein Extraction and Digestion

After SILAC labeling, cells were subjected to differential treatment: (1) complete medium with heavy arginine; (2) complete medium with light arginine; (3) arginine-free medium with 50 µM canavanine for 24 h; (4) AFM with 50 µM canavanine for 48 h. The treatments yielded different numbers of cells, due to differences in cell proliferation and survival rates (1−1.7 × 10^6^ cells). Cells were washed with ice-cold PBS three times and scraped off the plates in RIPA Buffer (250 µL per 10^6^ cells; see RIPA buffer composition above). The lysate was frozen on dry ice without preclearing. After thawing, the lysate was sonicated using a QSonica, Q125 sonicator probe homogenizer at 50% energy (1 min cycle; 15 s on and 15 s off). Protein from 320 µL (i.e., from around 1.2 × 10^6^ cells) was precipitated by the addition of 1280 µL of ice-cold acetone, incubated at −20 °C for 2 h and pelleted by centrifugation at 20,000× *g* for 30 min. The protein pellet was resuspended in 60 µL of 8 M urea and quantified with BCA (Pierce™ BCA Protein Assay Kit, ThermoFisher Scientific 23225). All samples had concentrations between 1.6–2 g/L. The samples were diluted 4-fold and digested at a 1:20 ratio of protein/Wako LysC (the protease was added at 1:40 twice with a 4 h interval). The peptides were desalted on an Oasis HLB 1 cc Vac Cartridge with 30 mg of sorbent as suggested by the manufacturer and speedvac-ed, and 1 µg of material was injected on the column.

### 2.12. LC-MS/MS Analysis

Samples were analyzed using a nanoAcquity UPLC system (Waters, Milford, MA, USA) coupled to an Orbitrap Fusion Lumos (ThermoFisher Scientific). Peptides were trapped on a Symmetry C18 5 μm, 180 μM × 20 mm precolumn (Waters, MA, USA) and desalted at 10 μL/min flow for 2 min with 0.1% TFA (trifluoracetic acid). The separation was performed on a nanoACQUITY CSH130 C18 1.7 µm, 75 µm × 250 mm at a 300 nL/min flow rate with 0.1% formic acid as Buffer A and 0.1% formic acid in acetonitrile as Buffer B. The gradient was 6–40% B over 75 min, 40–60% over 10 min and 60–100% over 15 min.

Eluted peptides were electrosprayed through an etched emitter [30] and analyzed using the Universal Method with a 60 s exclusion time (see the supplementary information S1 Experimental procedures: MS/MS method settings section). MS1 was performed in an orbitrap at a 120K resolution, and MS2, in ion trap rapid mode.

### 2.13. Proteomics Data Analysis

Data were analyzed in Proteome Discoverer in 2.4.035. A database search was performed with Sequest using the human UniProt database (retrieved on 06 June 2019). Methionine oxidation, ^13^C_6_,^15^N_4_-arginine and canavanine (user defined) were set as variable modifications; Cys carbamidomethylation was set as a fixed modification. The precursor and fragment mass tolerances were 3 ppm and 0.6 Da, respectively (we observed a large number of false positive canavanine identifications in samples where no canavanine labeling was performed at higher precursor mass tolerance). Peptides were validated with Percolator with a 0.01 posterior error probability (PEP) threshold. Peptide intensities were calculated with the Minora Feature Detector node. Top3 quantitation was performed only on samples with 48 h canavanine treatment in R using the dplyr package [31] as described previously [32]; only proteins with at least 2 proteotypic peptides were considered. GO (Gene Ontology) analysis was performed using the clusterProfiler R package [33]. The mass spectrometry proteomics data have been deposited to the ProteomeXchange Consortium via the PRIDE [34] partner repository with the dataset identifier PXD019044.

### 2.14. Theoretical Calculation of Necessary Mass Spectrometer Resolution

Calculations were performed essentially as described in [35] using the canavanine-containing peptides identified in this study. The atomic masses for ^1^H, ^12^C, ^13^C and ^16^O were taken as equal to 1.0078, 12, 13.0033 and 15.9949, respectively, from the CIAAW atomic weights of the elements (2019) [36]. The mass difference between the 3rd isotope of the unmodified peptide and 1st isotope of the canavanine-containing peptide is (13.0033–12) × 2 − (15.9949 − 12 − 1.0078 × 2) = 0.0273 or 27.3 mDa. The theoretical full width at 10% maximum (FWTM) peak height was computed for each identified peptide at the measured m/z and charge containing canavanine at 100 different resolutions (starting with 1 × 10^4^ and incrementing by 1 × 10^4^ until 1 × 10^6^) using Equation (1).
(1)$$FWTM=\;1.822\times \frac{m/z}{R\times \sqrt{\frac{200}{m/z}}}$$

The *FWTM* was then compared to the expected distance between the 3rd and the 1st isotope of the unmodified peptide, which was calculated as 27.3 mDa divided by the peptide charge.

### 2.15. Statistical Analysis

All experiments were run in triplicate and repeated at least three times. The results are expressed as means ± SD. Statistical analyses were performed using one way-ANOVA tests in the GraphPad Prism 8.4.3 software (San Diego, CA, USA). Statistical significance was defined as *p* < 0.05.

## 3. Results

### 3.1. Canavanine Treatment Decreases the Viability and Proliferative Potential of Human Glioblastoma Cells under Arginine Deprivation

First, we addressed the question whether the effects of canavanine on two human glioblastoma cell lines, U251MG and U87MG, which are known to differ in their proliferation, migration and invasiveness [37]. It was observed that canavanine strongly inhibited the viability of both cell lines in concentration- and time-dependent manners specifically under arginine deprivation (Figure 2a).

Although after 48 h of incubation in arginine-deficient medium, both cell lines could readily restore their proliferation when arginine was resupplemented. It was observed that preincubation with canavanine at concentrations above 50 µM essentially blocked such growth restoration (Figure 2b). Furthermore, the number of dead cells was noticeably increased after 48 h of treatment with 100 µM canavanine, which coincided with a progressive accumulation of the cleaved forms of PARP1 and caspase 3 after 48 h of treatment in both glioblastoma cell lines (Figure 3). Additionally, under canavanine co-treatment, a decreased level of the proapoptotic protein Bcl-2 was observed in U251MG cells (Appendix A). No significant differences in the level of the active cleaved form of caspase 9 in either cell line under all the examined conditions were detected (Appendix A).

These observations indicate that canavanine co-treatment, contrarily to arginine deprivation alone, induces caspase-dependent apoptotic cell death in the human U251MG and U87MG glioblastoma cell lines. Of note, we did not detect apoptotic markers in primary cultured rat glial cells under canavanine co-treatment (Appendix A).

### 3.2. Canavanine Profoundly Affects Morphology of U251MG and U87MG Cells under Arginine Deprivation

Many biological processes essential for both untransformed and cancer cells such as cell migration, morphogenesis, cytokinesis and endocytosis rely on the dynamics of the actin cytoskeleton [38,39]. Previously, we have also shown that arginine deprivation itself selectively destabilizes the actin cytoskeleton in glioma cells but not in primary rat glial cells [25]. Therefore, our next step was to examine the organization of actin filaments in the presence of 50 µM canavanine, a concentration near the IC50 for both examined cell lines. We noted that after 48 h of treatment in the absence of arginine, canavanine significantly affected the morphology of the examined U251MG and U87MG cells, especially the organization of the actin-rich structures of the leading edge such as lamellipodia and filopodia (Figure 4, solid arrows). This was accompanied by a decrease in the presence of filamentous actin as well as the appearance of bleb-like structures and actin-containing aggregates, which are indicative of progressing cell death. Both glioblastoma cell lines exhibited similar alterations. Importantly, there were no apparent changes either in the morphology or in the actin cytoskeleton organization in the glioblastoma cells treated with canavanine in CM or in the control rat glial cells subjected to combined treatment with canavanine and arginine deprivation (Figure 4).

It is known that actin filaments play an essential role in the control of both cellular and nuclear shape, and the geometry of the nucleus, in turn, affects cell proliferation, gene expression and protein synthesis [40,41,42]. A/C- and B-type lamins are major components of the nuclear lamina [43] that play an important role in the nucleo-cytoskeletal connection, and this interaction is essential for cell polarization, cell migration and cancer progression [41].

Staining for lamin B1 revealed abnormally shaped cell nuclei after 48 h of the culturing of U251MG cells, but not of U87MG ones, under arginine deprivation (Figure 5a, solid arrows). More pronounced changes in both glioblastoma cell lines were observed under canavanine co-treatment (Figure 5a, dotted arrows). Western blotting of the levels of nuclear lamins showed a decrease in lamin B1 in both glioblastoma cell lines and lamin B2 in U251MG cells under arginine deprivation in combination with canavanine treatment (Figure 5b).

We also observed fragmentation of lamins A/C and B1 in both U251MG and U87MG cells under prolonged co-treatment that could be also considered as one of the hallmarks of apoptosis (Figure 5b) [43]. Again, no substantial changes in the nuclear lamina pattern in analogously treated normal rat glial cells were observed (Appendix A).

Thus, our data show that combined treatment with arginine deprivation and canavanine evokes pronounced detrimental alterations in cytoskeleton and nuclear organization specifically in glioblastoma cells.

### 3.3. Lack of Arginine in Combination with Canavanine Profoundly Impairs Migration and Adhesion of Human Glioblastoma Cell Lines

Cell migration and invasion play a key role in cancer metastasis [44]. We previously demonstrated that arginine deficiency specifically impaired the motility and adhesive interactions of human glioblastoma and melanoma cells [25,45]. The above-described exacerbated alterations of cell morphology observed under canavanine co-treatment could be associated with changes in cell adhesion and motility. Therefore, we analyzed the cell migration of U251MG and U87MG glioblastoma cells treated with canavanine under arginine deprivation using the Transwell™ system [46]. While arginine deprivation itself profoundly decreased migration by ~50%, almost complete inhibition of cell penetration through the porous membrane after 48 h of canavanine co-treatment was observed for both glioblastoma cell lines (Figure 6).

Cell adhesion to the extracellular matrix is a highly dynamic process that plays a crucial role in the regulation of cell proliferation and growth, cell motility and gene expression [47,48,49]. We analyzed the focal adhesion structures in glioblastoma cells using immunostaining for an integrin-associated linker protein, vinculin. We observed that canavanine augmented alterations in the focal contact morphology evoked by arginine deprivation in both studied cancer cell lines (Figure 7a). This was more pronounced for U251MG cells, where vinculin-stained adhesive structures were smaller and more dispersed. Additionally, changes in focal contact structures were also noticeable in a control primary rat glial cell under canavanine co-treatment, although they were not as pronounced as in both cancer cell lines (Appendix A).

Next, we analyzed cell adhesion by the assessment of the cell-matrix interactions of glioblastoma cells with a surface covered with collagen I, one of the main components of the extracellular matrix. We observed that under arginine deprivation, canavanine caused a dramatic decrease (by ~75%) in cell adherence to the surface of both glioblastoma cell lines (Figure 7b). Of note, arginine deprivation itself practically did not evoke a negative effect on adhesion to collagen I.

Despite such evident changes in adhesive structures and cell-surface adhesion, the levels of proteins involved in cell adhesion such as vinculin, talin and tensin 2 did not change substantially with respect to those in control cells (Figure 7c and Appendix A). Since focal adhesion kinase (FAK) is a key regulator of adhesive structure organization, we next assessed its levels as well as the level of its phosphorylated (active) form (p-FAK, Tyr397) in the examined conditions. We observed a substantial reduction of the p-FAK level under arginine deprivation in the absence of canavanine (Figure 7c). Combined treatment with canavanine for 48 h augmented the decrease only in U251MG cells. Interestingly, the presence of canavanine in the complete medium also led to a decrease in FAK activity (Figure 7c). Similar, but less dramatic, effects were observed in primary rat glial cells (Appendix A).

It is known that actin-cytoskeleton organization, cell morphology and survival as well as cell adhesion are under the control of Akt kinase [50,51,52,53]. Therefore, we examined the activation level of this kinase by analysis of its phosphorylation at S473. As shown in Appendix A**,** 48 h of incubation in arginine-free conditions with canavanine treatment led to a ~50% decrease in Akt activation (phosphorylation at Ser473) in both glioblastoma cell lines. Moreover, 48 h of canavanine treatment led to a decrease in total Akt protein in U251MG cells.

Overall, our data suggest that canavanine effectively augments the negative effects of arginine deprivation on glioblastoma cells’ heterotypic adhesion.

### 3.4. Proteomics Analysis of Canavanine Incorporation into Proteins in U251MG Glioblastoma Cells

While previous reports have suggested that canavanine gets incorporated into polypeptides during translation [13,27], the direct observation of canavanine in proteins by mass spectrometry and the extent of its incorporation or whether it is more likely to get incorporated as part of a motif have not been reported. The challenge is that Arg-to-canavanine substitution would result in a 1.9793 shift (16O–12C–1H × 2), which is very close to the mass difference of +2.0066 Da between the third and the first isotopes of an unmodified peptide, 2 × (13C–12C). Resolving this difference of 27.3 mDa on an Orbitrap LUMOS in MS1 would require operating at a 240K resolution for half of the peptides and 540K for 95% of the peptides (Figure 8a, see Materials and Methods for a description of the calculation). Such a high resolution is currently not attainable on this platform but would also be impractical for operation, due to the very long scan times on the majority of Orbitrap platforms [54]. Moreover, even if the unmodified and canavanine-containing peptides were resolved at the MS1 level, it would be impossible to generate their separate MS2 spectra due to quadrupole co-isolation. Hence, we labeled arginine (Arg) with ^13^C_6_^15^N_4_ to shift the masses of Arg-containing peptides by +10 Da to avoid the overlap with canavanine-containing peptides. Unlabeled arginine was termed as light Arg, and the labeled one was termed as heavy Arg. We also used endoproteinase LysC as the only protease, since the efficiency of trypsin for canavanine is currently unclear. Treatment with LysC generates longer and hence more hydrophobic peptides, and we also optimized the LC gradient for a higher final concentration of ACN.

To demonstrate canavanine incorporation into the proteins, we labelled U251MG cells with heavy Arg for seven passages and then transferred them to one of the following media: (i) AFM with 0.1 mM light Arg, (ii) AFM with 0.1 mM heavy Arg, and (iii) AFM with 50 µM canavanine. Cells were harvested, and their proteome was analyzed after 48 h. An additional time point at 24 h was added for the AFM with 50 µM canavanine. Each sample was analyzed in biological triplicate, and we calculated the incorporation of light Arg, heavy Arg and canavanine as proportions of identifications and signals (Table 1, only Arg-containing peptides were considered). Assuming that canavanine incorporation does not significantly affect ionization efficiency, the proportion of the signal can be interpreted as the proportion of canavanine at all residues normally occupied by arginine. We observed that ~12.6% of all Arg sites were occupied by canavanine after culturing cells in media with 50 µM canavanine for 48 h. In comparison, culturing with 100 µM light Arg for the same period of time resulted in 68.3% incorporation.

In total, we identified 1589 peptide sequences containing canavanine (see Appendix A) corresponding to 1980 canavanine sites. We investigated whether canavanine incorporation was more or less likely in the context of other amino acids using the motif-x software [55]. We used the sequences of canavanine-containing peptides as the foreground set and used all the identified Arg (or canavanine)-containing sequences as the background set (i.e., we compared the sequences of canavanine-containing peptides and all Arg/canavanine peptides). We looked at five amino acids at both sides of each heavy Arg (or canavanine) site. We identified no enriched motifs at the *p*-value of 1 × 10^−6^ (recommended settings in the motif-x software) (Appendix A).

We also tested whether specific functional categories were enriched among the proteins labeled with canavanine (using all the identified proteins as a background) in samples treated with canavanine for 48 h using the clusterProfiler R package [33]. While there was a significant enrichment of many GO terms, especially those related to cell adhesion and the unfolded protein response (see Appendix A for the list of proteins used in the GO analysis, and Appendix A and Appendix A for the set of enriched categories), the implications of this enrichment are unclear. This could be interpreted as canavanine being preferentially incorporated into these proteins or these proteins being preferentially synthesized under canavanine treatment. At the same time, the group of canavanine-labeled proteins was also significantly more abundant (Appendix A), and canavanine-labeled peptides were generally less intense than their heavy-Arg counterparts (see below). This could mean that peptides containing canavanine fall below the limit of detection in less-abundant proteins, resulting in the high abundance of proteins enriched in the canavanine-labeled set.

The sequences of 1081 peptides had two variants identified within a single run: a variant with a single incorporated canavanine and a variant with all heavy Arg (Appendix A). We used this peptide set to investigate the effect of canavanine incorporation on the peptide charge and retention time, since a significant change in these properties would allow the analysis of canavanine-containing peptides in label-free proteomics data. Interestingly, we did not find a significant effect of canavanine on either of these properties (see Figure 8b,c), validating the necessity of heavy-Arg labeling. The median ratio of the intensities for heavy-Arg to canavanine-containing peptides was 7.5 (see Figure 8d).

As a final illustration of our strategy, we observed a small subset of peptides that had all three variants (i.e., heavy Arg, light Arg and canavanine) present at a certain Arg position, due to the incomplete heavy-Arg labeling prior to canavanine treatment. Appendix A demonstrates the elution profiles and spectra for the three variants for one such peptide. As expected, the canavanine and light-Arg peptide variants were not resolved at the MS1 level at a 120 K resolution.

### 3.5. Canavanine Enhances ER Stress Evoked by Arginine Deprivation

Since arginine deficiency has been shown to induce endoplasmic reticulum (ER) stress in many cancer cell lines [56] and our mass spec data showed a significant enrichment of GO terms related to the unfolded protein response, our next goal was to analyze the effect of canavanine on this process in human glioblastoma cells. We performed immunofluorescence staining for a marker of ER stress, namely, the ER chaperone 78-kDa glucose-regulated protein (GRP78) [57,58]. We noticed an accumulation of GRP78 within the ER of both analyzed glioblastoma cell lines after 48 h of incubation under arginine deprivation in the presence of canavanine (Figure 9a, see arrows). These observations were confirmed by Western Blot analysis, which showed a significant increase in the level of GRP78 in cell lysates (Figure 9b). On the other hand, canavanine co-treatment only slightly increased the level of GRP78 in primary rat glial cells under analogous conditions (Appendix A).

In order to visualize the ER tubular network, we performed staining with the ER-specific dye, ER Tracker Blue/White DPX, which is retained within the ER lumen. As shown in Appendix A, the organization of the ER was changed in both arginine-deprived glioblastoma cell lines, and numerous ER tracker-stained vesicles (possibly autophagosomes) became visible. Additionally, 48 h of co-treatment with canavanine caused a complete disintegration of the ER network (Appendix A).

It is known that the accumulation of unfolded or misfolded proteins within the ER lumen activates unfolded protein response pathways implicated in protective/adaptive responses and in the promotion of apoptosis [59,60]. We examined the activation status of several proteins involved in the regulation of protein synthesis in our experimental conditions. As shown in Appendix A, already 4 h of culture with 50 µM canavanine in arginine-free medium evoked pronounced changes in the phosphorylation status of proteins involved in protein synthesis such as 4EBP1 (mTOR pathway) and eIF2α (GCN2 pathway) in both glioblastoma cell lines. Prolonged arginine deprivation or combined treatment further promoted the hyperphosphorylation of eIF2α and dephosphorylation of ribosomal protein S6 and 4EBP1, and canavanine only partially affected the processes leading to the inhibition of global protein translation (Figure 10a). Of note, canavanine also upregulated the dephosphorylation of 4EBP1, even in complete medium, in U87MG cells, a cell line that is slightly more sensitive to this toxic arginine analogue (Figure 1).

Additionally, the level of a cAMP-dependent transcription factor, ATF4, a marker of ER stress-induced apoptosis, was dramatically increased under co-treatment with canavanine, in comparison to that with single arginine deprivation as a monotreatment (Figure 10b). Interestingly, canavanine under arginine-free conditions induced the formation of an additional ~38 kDa band, probably ATF4 isoform CRA_a, which has been described in the literature [61].

Canavanine also evidently augmented the proapoptotic signaling pathway activated by arginine deprivation as evidenced by the increased phosphorylation of stress-activated protein kinase SAPK/JNK and mitogen-activated protein kinase p38 in glioblastoma cells (Figure 10b). Interestingly, the increase in the level of the proapoptotic transcription factor CHOP observed under arginine deprivation was not further elevated by the presence of canavanine in either glioblastoma cell line (Figure 10b).

Analysis of the rat glial cells’ response to canavanine under arginine deprivation revealed the dephosphorylation of the S6 protein and a weak activation of proapoptotic markers such as ATF-4, p-SAPK/JNK and p-p38 after 48 h of co-treatment (Appendix A).

These data indicate that canavanine significantly enhances ER stress and profoundly promotes proapoptotic responses selectively in malignant U251MG and U87MG glioblastoma cells.

### 3.6. Canavanine Affects Mitochondria under Arginine Deficiency

It is known that mitochondrial dynamics can be significantly disturbed under stress conditions, and their adaptations are crucial for many cellular processes [62,63]. Additionally, the accumulation of misfolded or damaged proteins in mitochondria under metabolic stress can lead to organelle dysfunction. In order to examine whether canavanine in arginine-deprived conditions could also affect mitochondria, we performed immunocytochemical staining for heat shock protein glucose-regulated protein 75 (GRP75/mortalin), essential for maintaining the ER–mitochondrial contacts (Figure 11a) [64,65]. Under arginine deprivation, we observed a long, apparently hyperfused mitochondrial network, which was especially visible in U251MG cells (Figure 11a, solid arrows). Additionally, canavanine evoked dramatic changes, as the mitochondrial network became disrupted and aggregate-like structures were visible in both cell lines (Figure 11a, dotted arrows). Western blot analysis showed a decrease in mortalin levels in both cell types treated with canavanine in the absence of arginine for 48 h, but statistical significance was only found for U251MG cells (Figure 11b,c). Moreover, a concomitant significant increase in the heat shock proteins HSP70 and HSP60 involved in mitochondrial stress was observed in glioblastoma cells under canavanine co-treatment (Appendix A).

It should be also emphasized that canavanine under arginine deprivation caused a decrease in the phosphorylation of a catalytic α subunit of AMP-dependent kinase, AMPK, a known cellular energy sensor, indicative of the inhibition of the kinase’s activity (Figure 12).

These data show that the arginine analogue canavanine, under arginine deprivation, induces the unfolded protein response in mitochondria and inhibits the activation of AMPK, responsible for energy homeostasis.

## 4. Discussion

Glioblastoma is one of the most frequent and aggressive forms of primary brain tumors that develops from glial cells such as astrocytes and oligodendrocytes [66]. Glioblastomas are difficult to eliminate surgically and to treat in general due to their highly infiltrative and invasive nature [66]. Moreover, the blood–brain barrier prevents the delivery of many medications from the bloodstream to glioblastoma sites. The average survival time for glioblastoma patients after diagnosis is only 15–18 months, and the 5-year survival rate is only 10% [67]. Today, clinical trials are exploring new, potentially more efficient antiglioblastoma approaches in chemotherapy, radiation therapy, immunotherapy or their combinations [67]. We and others proposed that anticancer therapy based on single-amino-acid arginine deprivation that utilizes recombinant arginine-degrading enzymes could be applied as an antiglioblastoma treatment [1,24,25]. This metabolic approach has already achieved significant progress in cell culture and animal studies, and in clinical trials aimed at a growing number of cancer types [68]. Although the feasibility of this approach for the treatment of various cancers, primarily those with defects in arginine anabolism, has been demonstrated [67,68], we have found that the strong antitumor effect of arginine starvation observed in standard in vitro monolayer cell cultures does not fully translate into 3D cultures [69]. We and others proposed, therefore, that rationally designed combinational modalities are needed for the approach to successfully proceed into clinical use [1,15].

As we previously reported, arginine deprivation selectively and profoundly affected human glioblastoma cell morphology and migration, but these effects did not lead to glioma cell death and were fully reversible after arginine resupplementation [25]. In this study, we aimed to examine whether a proteomimetic plant arginine analogue, canavanine, could augment the effect of arginine deprivation on two human glioblastoma cell lines, U251MG and U87MG. Additionally, we addressed the molecular mechanisms behind canavanine’s effects on human glioblastoma cells.

We revealed that the combination treatment with canavanine had the strong time- and concentration-dependent cytotoxic effects on both examined human glioblastoma cell lines. The effects were irreversible after 48 h of co-treatment with 50 µM canavanine, in contrast to in the arginine-deprivation-only condition, in which resupplementation with arginine fully restored cell growth. These observations support our notion that the combination of arginine starvation with canavanine could be considered as a promising new modality for antiglioblastoma treatment independent of the blood–brain barrier.

It is known that tumor invasiveness and aggressiveness are highly dependent on malignant cell motility. A negative impact of arginine deprivation as a monotreatment on the motility of glioblastoma, melanoma and colon carcinoma cells was observed [25,45,70]. This phenomenon in glioblastoma cells was most probably evoked by the decrease in the content of the positively charged arginylated form of β-actin and polymerized F-actin [25]. Herein, using several independent techniques, we showed that canavanine significantly augmented the effects of arginine deprivation on the morphology and migration of glioblastoma cells. Dramatic changes in the actin cytoskeleton organization, as well as changes in cell polarity and adhesion, such as the substantial loss of lamellipodia and defects in adhesive structures, essentially abrogated cancer cell migration after 48 h of co-treatment. The co-treatment significantly affected the FAK/Akt/Bcl-2 signaling pathway by the deactivation of FAK and Akt kinases, the main players in the regulation of cell migration and survival [71]. At the same time, canavanine did not evoke significant effects on the morphology and adhesion of normal rat glial cells deprived of arginine.

Alterations in the actin cytoskeleton’s organization are also known to affect the nuclear architecture [72]. The nuclear lamina, a key structural element of the nucleus, is a complex protein network that provides structural support to the nuclear envelope and serves as the structural link between the nucleo- and cytoskeleton [42,73,74]. Upon canavanine co-treatment, we observed specific disturbances in the nuclear lamina, which could affect the mechanical stability of the nucleus, genome organization and signal transduction. Furthermore, based on numerous reports, it is plausible that the here-observed fragmentation of lamins A/C and B1 is associated with the initiation of apoptotic cell death [75,76].

In order to reveal the mechanism(s) behind the observed effects, we examined whether they could result from canavanine incorporation into polypeptide chains under arginine deprivation. Previously, it was shown with the use of cycloheximide, an inhibitor of protein biosynthesis, that canavanine incorporation into nascent proteins could be a cause its observed cytotoxic effects on several cancer cell lines such as keratinocytic carcinoma A431, lung adenocarcinoma A549, hepatocellular carcinoma HepG2, and pancreatic carcinoma MIA PaCa-2 [13]. Of note, we have also demonstrated that canavanine is much more cytotoxic towards malignant cells of different organ origin than towards pseudonormal cells [6,13]. Moreover, we showed that, in vitro, it is a rather weak substrate for recombinant human arginase [13,77]. Of note, canavanine’s proteomimetic capability has been suggested as a main contributing factor in toxicologic diseases, for which data are already available in proteomics repositories [34]. We were first to observe and directly quantify canavanine incorporation into polypeptide chains in the examined U251MG cells with the use of LC-MS/MS analysis. Importantly, we observed that the analysis of label-free data from proteomics repositories for canavanine incorporation is not feasible since the MS1 signals for unmodified and canavanine-containing peptides would overlap. To address this challenge, we successfully employed the method of SILAC arginine labeling [29] for our model of aggressive and chemotherapy-resistant glioblastoma U251MG cells. Our analysis revealed that canavanine in the absence of arginine was indeed readily incorporated into the polypeptide chains at rates of up to ~13%. Of note, we did not observe a motif preference for canavanine incorporation, and while certain GO categories were enriched among the proteins labeled with canavanine, the biological implications of this are not clear at the moment.

It is well established that amino acid starvation, in general, triggers the transcriptional amino acid response pathway and inhibits mTOR signaling [78]. It was also demonstrated that single-amino-acid arginine deprivation halts global protein synthesis but activates the machinery of a stress response [56,79]. Therefore, it was interesting—knowing that in, the examined conditions, canavanine is indeed built into the proteins—to address the effects of canavanine on these mechanisms. We observed that canavanine co-treatment had complex activating effects on the ER stress response, with the involvement of several signal transduction pathways. We observed both the suppression of mTOR activity and activation of the PERK signaling cascade of the stress response, causing an arrest of protein synthesis. Moreover, it is known that the hyperphosphorylation of eIF2α significantly promotes the synthesis of the activating transcription factor 4 (ATF4), and this alters the pattern of gene expression, for example, of CHOP, a protein involved in the induction of apoptosis [80]. Furthermore, we observed that canavanine under arginine deficiency stimulated the mitogen-activated protein kinase (MAPK) pathway by the dramatic activation of the SAPK/JNK and p38 kinases, which play important roles in inflammatory gene expression and apoptosis [80,81].

The ER compartment is known to be associated with mitochondria, which play an essential role in the cell’s energetic metabolism and calcium storage, and in the control of cell death [82,83]. We demonstrated that canavanine under arginine deprivation affected mitochondrial morphology and AMPK signaling, thus pointing at possible defects in energy metabolism.

Summarizing, our data indicate that under arginine deprivation, the arginine analogue canavanine has multifaceted effects on glioblastoma cells that, in the end, promote cell death. Moreover, these effects predominantly concerned transformed glioblastoma cells but not normal glial cells. We are aware, however, that this type of treatment, like many other types of anticancer therapy, could be somewhat stressful for healthy cells/tissues. The selective cytotoxic effect of canavanine on cancer cells is due to its more efficient uptake by these cells, which unlike normal cells, have defects in arginine metabolism.

Our findings thus provide strong mechanistic grounds for considering the co-treatment as a possible novel strategy for the development of a specific and effective antiglioblastoma metabolic therapy independent of the blood–brain barrier. To our knowledge, no similar approaches, in particular, against glioblastoma, are considered in clinics. However, numerous clinical studies on arginine-degrading enzymes, or their combination with certain chemotherapy drugs, are in progress for several tumor entities [7,10,84].

## Figures and Tables

**Figure 1 cells-09-02217-f001:**
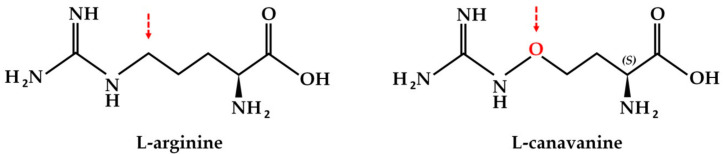
Chemical structures of l-arginine and l-canavanine.

**Figure 2 cells-09-02217-f002:**
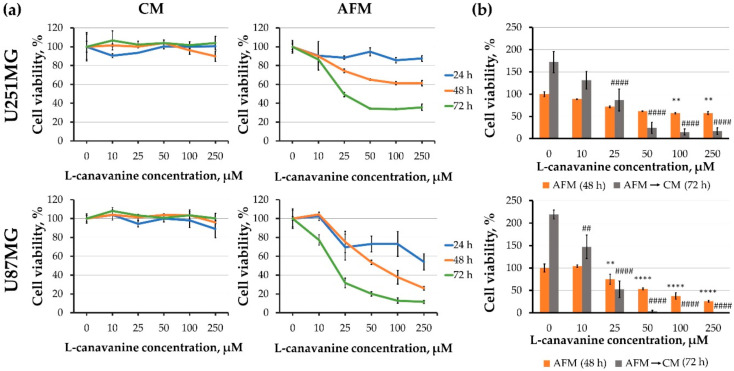
Effect of canavanine on glioblastoma cell viability and ability to restore growth upon arginine resupplementation. (**a**) MTS assay for U251MG and U87MG human glioblastoma cells treated for up to 72 h with increasing concentrations of canavanine (CAV) in the complete (CM) or arginine-free (AFM) media. (**b**) After 48 h of treatment, the medium was changed for arginine-containing complete medium (CM), and the cells were cultured for an additional 72 h. Then, growth restoration was assessed using the MTS test. Graphs represent mean values ± SD from three independent experiments. ** *p* < 0.01, **** *p* < 0.0001 relative to arginine-free medium (AFM) (control, 100%); ^##^
*p* < 0.01, ^####^
*p* < 0.0001 relative to control cells after resupplementation.

**Figure 3 cells-09-02217-f003:**
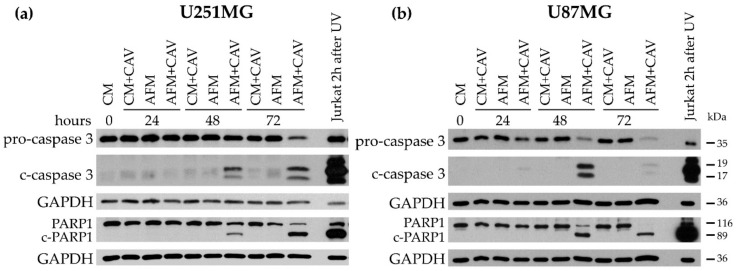
Assessment of the levels of the markers of apoptotic cell death in U251MG (**a**) and U87MG (**b**) cells under 100 µM canavanine treatment in complete or arginine-free medium. GAPDH was used as a protein loading control. UV-irradiated Jurkat cells were used as a positive control.

**Figure 4 cells-09-02217-f004:**
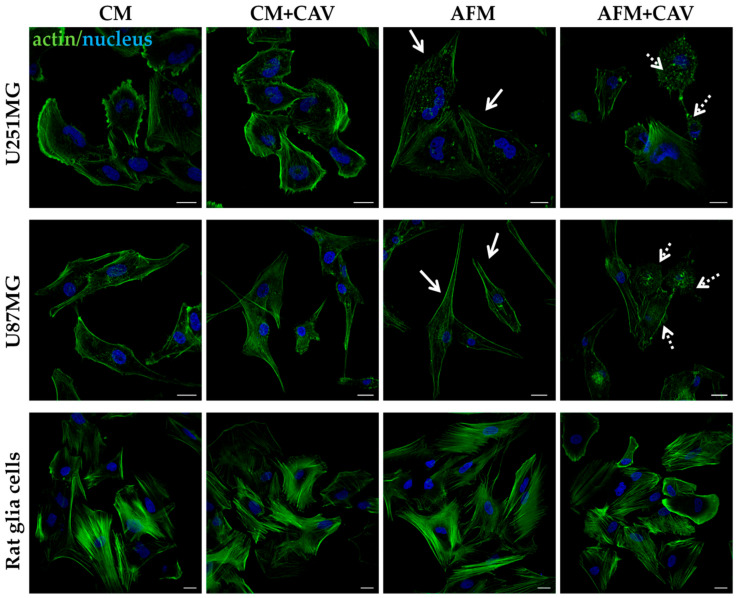
Organization of the actin cytoskeleton, probed with Alexa Fluor 488-conjugated phalloidin, of U251MG and U87MG human glioblastoma cells as well as normal rat glial cells. Cells were cultured for 48 h under experimental conditions with or without 50 µM canavanine. Untreated cells (CM) served as a control. Nuclei were labeled with DAPI. Solid arrows indicate altered leading-edge morphology in cancer cells starved for arginine, and dotted arrows indicate actin-rich aggregates and blebs elicited by canavanine. Bars, 20 µm.

**Figure 5 cells-09-02217-f005:**
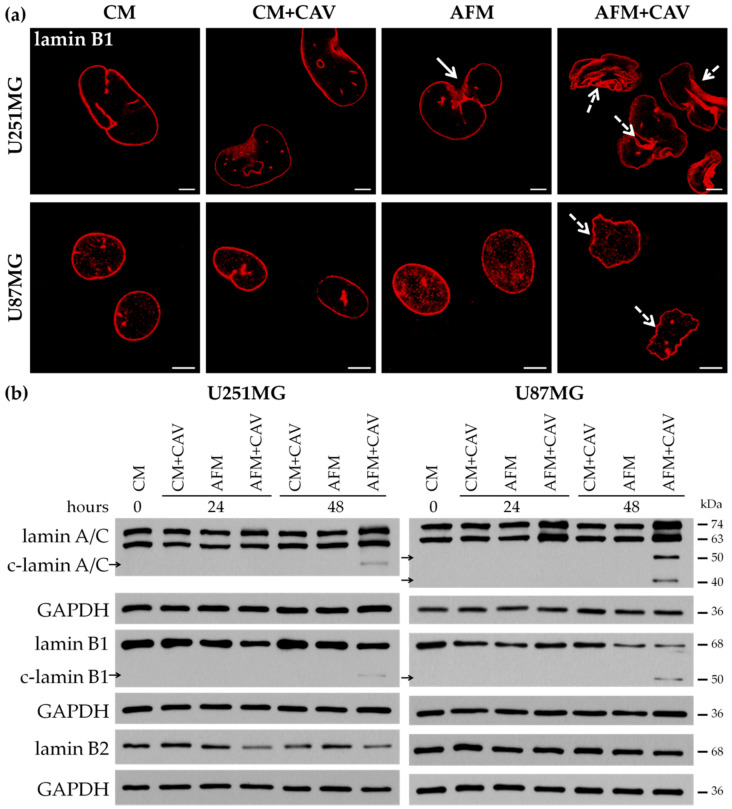
Analysis of nuclear lamina in U251MG and U87MG cells undergoing combined treatment of arginine deprivation with 50 µM canavanine. (**a**) Immunofluorescence analysis of lamin B1. Solid arrows indicate abnormal nuclei under arginine deprivation; dotted arrows point to misshaped cell nuclei under co-treatment. Bars, 5 µm. (**b**) Western blots of nuclear envelope proteins lamin A/C and lamin B1 and their cleaved forms (c-lamin A/C and c-lamin B1) and lamin B2. GAPDH was used as a protein loading control.

**Figure 6 cells-09-02217-f006:**
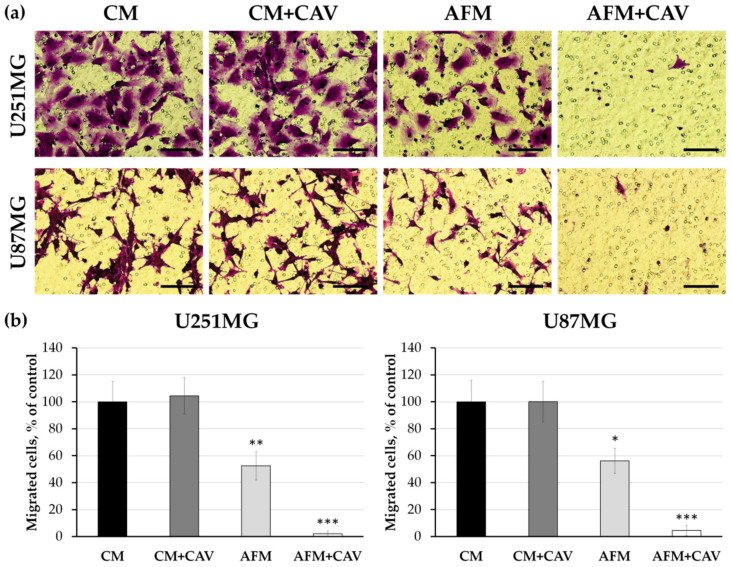
Effect of canavanine on human glioblastoma cell migration. (**a**) Representative images of migrated cells after the 48 h of co-treatment with 50 µM canavanine under CM and AFM conditions. Bars, 100 μm. (**b**) Quantification of migrated cells. Graphs represent mean values ± SD from three independent experiments. * *p* < 0.05, ** *p* < 0.01, *** *p* < 0.001 relative to CM (control, 100%).

**Figure 7 cells-09-02217-f007:**
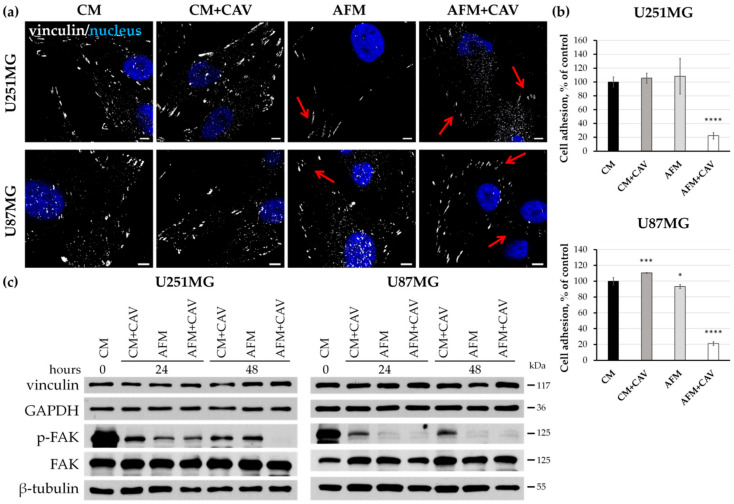
Effects of canavanine on adhesion of human glioblastoma cells. (**a**) Immunocytochemical staining for vinculin. Nuclei were labeled with DAPI. Cells were treated for 48 h under indicated experimental conditions under CM and AFM, with or without 50 µM canavanine. Bars, 10 µm. Arrows point to altered focal adhesion contacts. (**b**) Cancer cell adhesion to collagen I after the 48 h of treatment in CM and AFM with or without CAV. Graphs represent mean values ± SD from three independent experiments. * *p* < 0.05, *** *p* < 0.001, **** *p* < 0.0001 relative to CM (control, 100%). (**c**) Immunoblotting of proteins involved in cell adhesion and migration. GAPDH and β-tubulin were used as protein loading controls.

**Figure 8 cells-09-02217-f008:**
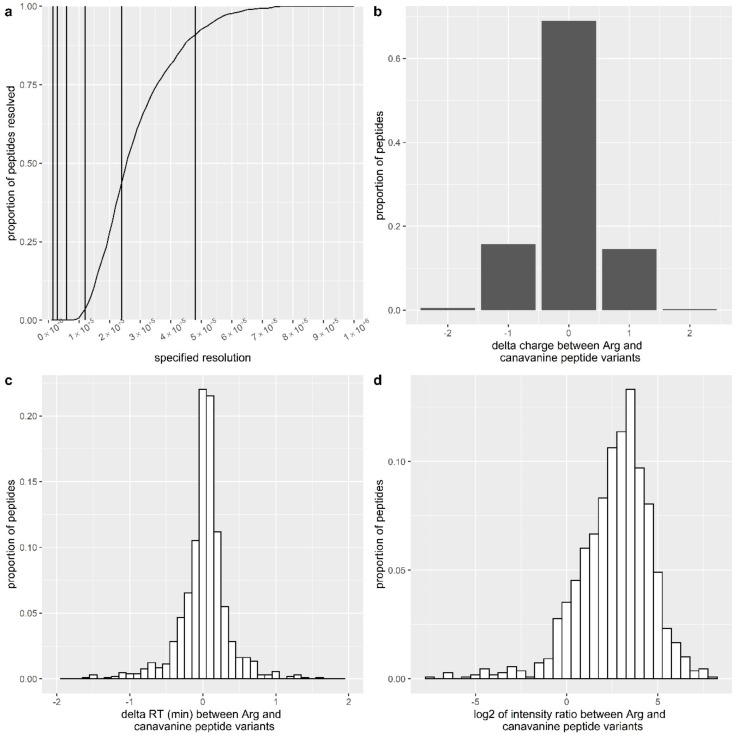
Proteomics of canavanine-bearing peptides. (**a**) Proportion of peptide variants with light Arg and canavanine resolved at different instrument resolutions (vertical lines indicate resolution settings available on Orbitrap Lumos). (**b**–**d**) Effect of canavanine incorporation on peptide charge state, retention time and intensity.

**Figure 9 cells-09-02217-f009:**
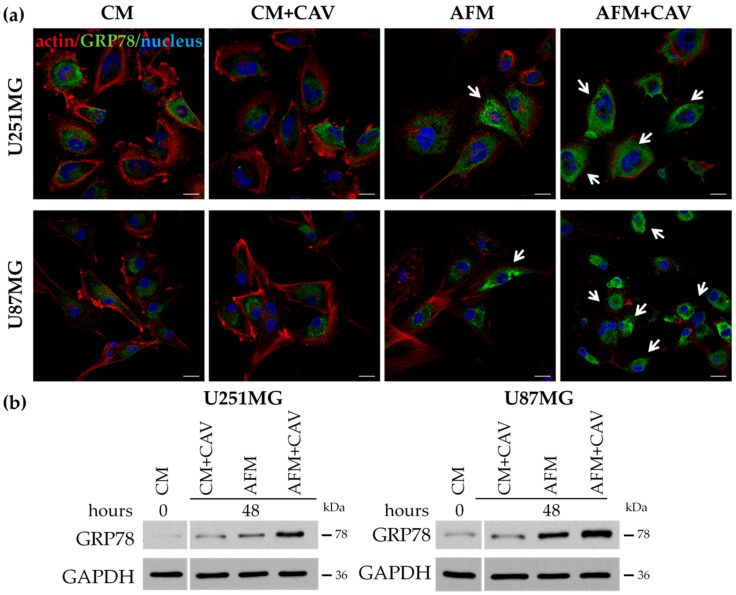
Canavanine (50 µM) enhances endoplasmic reticulum (ER) stress during arginine deprivation in U251MG and U87MG human glioblastoma cells. (**a**) Immunostaining for GRP78 and actin filaments of the cells subjected to 48 h of canavanine co-treatment under arginine starvation. Nuclei were stained with DAPI. Bars, 20 µm. Arrows point to cells with increased GRP78 fluorescence. (**b**) Western blot analysis of GRP78 protein level. GAPDH was used as a protein loading control.

**Figure 10 cells-09-02217-f010:**
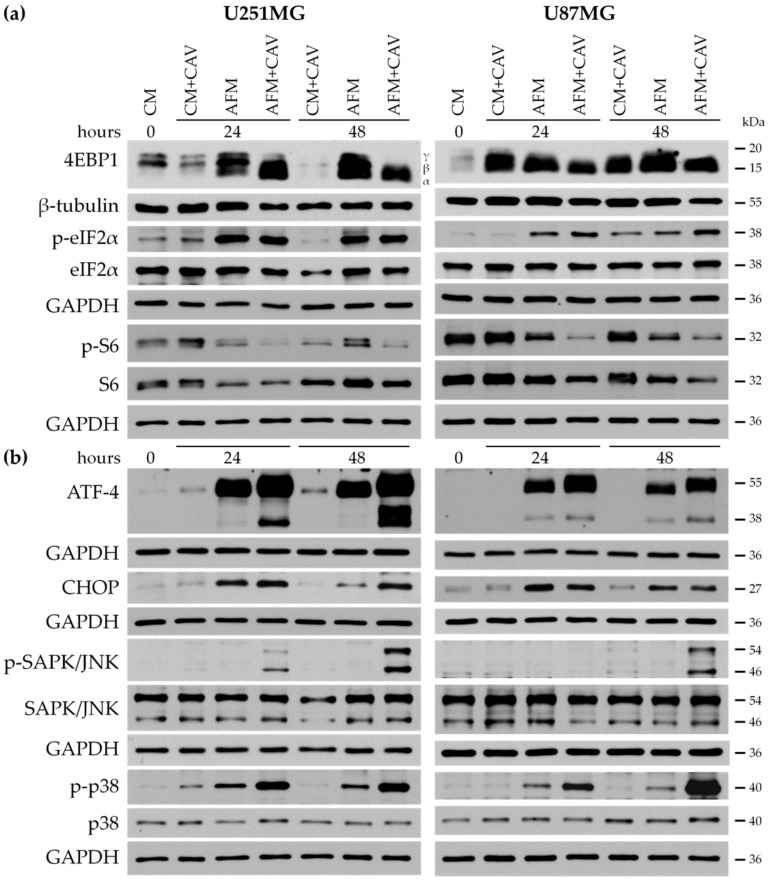
Western blot analysis of selected molecular markers of ER stress and cells’ adaptive responses. (**a**) Proteins involved in protein synthesis. (**b**) Proapoptotic markers. Cells were incubated for up to 48 h under experimental conditions with or without 50 µM canavanine. GAPDH was used as a protein loading control.

**Figure 11 cells-09-02217-f011:**
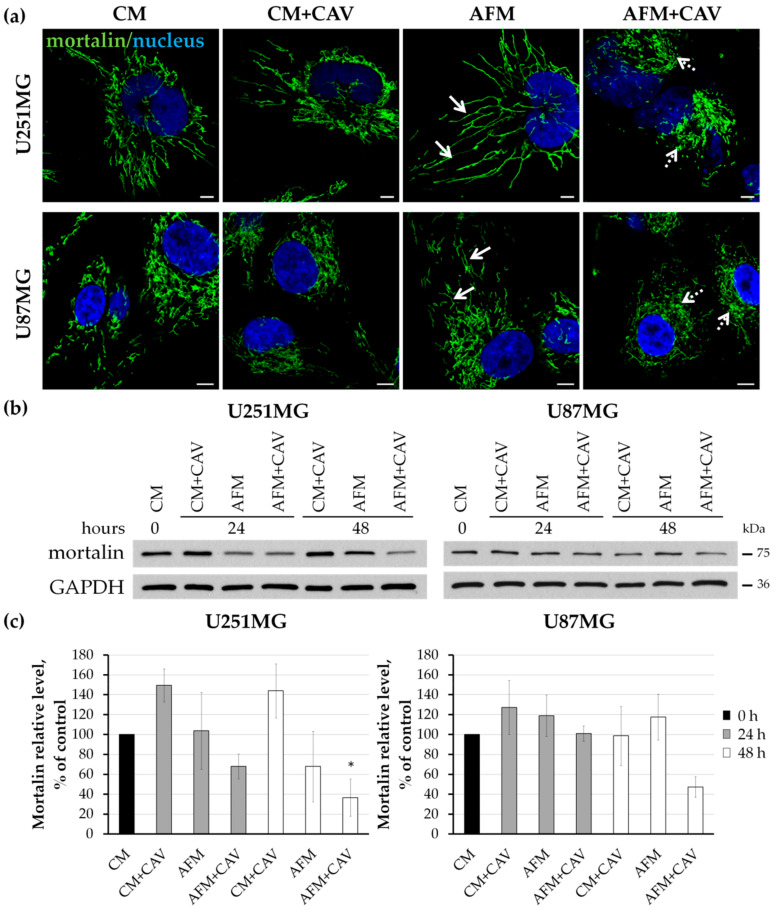
Assessment of mitochondrial organization in human U251MG and U87MG glioblastoma cells treated with 50 µM canavanine under CM and AFM conditions. (**a**) Immunostaining for mortalin. Nuclei were labeled with DAPI. Cells were treated for 48 h under experimental conditions. Bars, 5 µm. Arrows indicate hyperconnected mitochondria, and dotted arrows point to aggregate-like mitochondria. (**b**) Western blot analysis of mortalin levels. GAPDH was used as a protein loading control. (**c**) Densitometry of mortalin protein levels. Graphs represent mean values ± SD from three independent experiments. * *p* < 0.05 relative to CM (control, 100%).

**Figure 12 cells-09-02217-f012:**
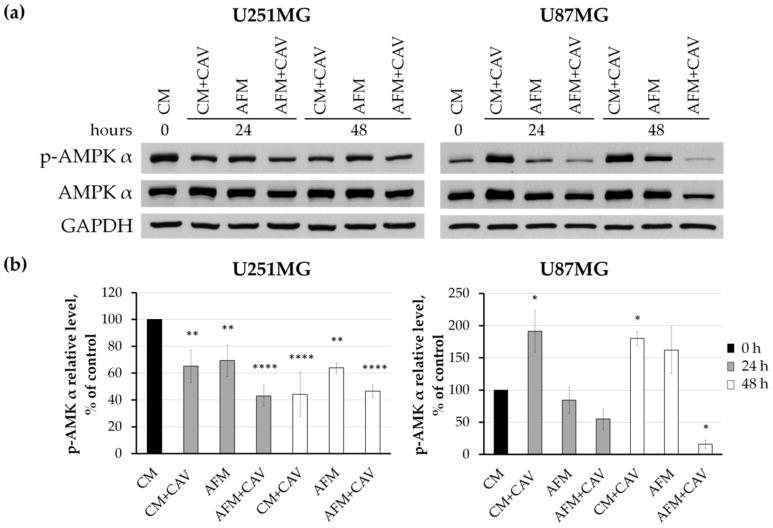
Analysis of the level of 5′-AMP-activated protein kinase AMPK α subunit. (**a**) Western blotting of AMPK α subunit and its phosphorylated form (*p*-AMPK α). GAPDH served as the internal loading control. (**b**) Densitometry of AMPK level. Graphs represent mean values ± SD from three independent experiments. * *p* < 0.05, ** *p*< 0.01, **** *p* < 0.0001 relative to CM (control, 100%).

**Table 1 cells-09-02217-t001:** Incorporation of arginine (Arg) and canavanine (CAV) into proteins of U251MG cells under examined conditions.

	ArgH, 48 h	ArgL, 48 h	CAV, 24 h	CAV, 48 h
% of identifications				
ArgH	97.16 ± 0.19	33.85 ± 0.86	83.59 ± 0.47	76.91 ± 0.26
ArgL	2.78 ± 0.2	65.94 ± 0.83	6.48 ± 0.44	6.3 ± 0.23
CAV	0.06 ± 0.03	0.22 ± 0.04	9.93 ± 0.28	16.79 ± 0.46
% of total intensity				
ArgH	98.72 ± 0.07	31.59 ± 0.95	90.09 ± 0.39	83.48 ± 0.38
ArgL	1.24 ± 0.06	68.34 ± 0.95	4.08 ± 0.38	3.88 ± 0.63
CAV	0.04 ± 0	0.06 ± 0.01	5.83 ± 0.11	12.63 ± 0.39

Incorporation presented as proportions of identifications (top part of the table) and proportions of intensity (bottom part of the table) ± SD. Three replicates were recorded per condition. ArgH, heavy arginine; ArgL, light arginine.

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
