# Peer review of "Combinatory Treatment of Canavanine and Arginine Deprivation Efficiently Targets Human Glioblastoma Cells via Pleiotropic Mechanisms"

_cells, 2020, doi:10.3390/cells9102217_

Round 1

Reviewer 1 Report

I read with interest and pleasure the manuscript submitted by Karatsai et al. It is a well structured paper with a great number of used methods and clearly exposed results.

A possible critical aspect may be represented by the fact that the administration of canavanine plus arginine deprivation may lead to a slight ER stress and to a weak activation of pro-apoptotic factors after 48 h of treatment in normal glial cells. However, it is also to consider that these cells are from rat brain and it would be interesting to know how it could be the behavior of normal human astrocytes. Anyways, the overall data presented are sufficient to affirm that the proposed co-treatment has a valid anti-GBM effect  and, likely, few negative effects on normal glia.

Reviewer 2 Report

Interesting experimental study regarding the metabolism of glioblastoma cells.

How could this be incorporated as a therapy?

Could you comment on similar therapy approaches?
